# Acoustic and perceptual impact of face masks on speech: A scoping review

**Gursharan Badh[1]ஃ, Thea Knowles [2]ஃ***

**1** Department of Communicative Disorders & Sciences, University at Buffalo, Buffalo, NY, United States of America, **2** Department of Communicative Sciences & Disorders, Michigan State University, East Lansing, MI, United States of America

ஃ These authors contributed equally to this work.
* thea@msu.edu

**Data Availability Statement:** https://github.com/thealk/Speech-In-Masks_scoping_review.

**Funding:** The authors received no specific funding for this work.

## Abstract

During the COVID-19 pandemic, personal protective equipment such as facial masks and coverings were mandated all over the globe to protect against the virus. Although the primary aim of wearing face masks is to protect against viral transmission, they pose a potential burden on communication. The purpose of this scoping review was to identify the state of the evidence of the effect of facial coverings on acoustic and perceptual speech outcomes. The scoping review followed the framework created by Arksey & O'Malley (2005) and the Preferred Reporting Items for Systematic Reviews and Meta-Analyses Extension for Scoping Reviews guidelines (PRISMA-ScR; Tricco et al., 2018). The search was completed in May 2021 across the following databases: PubMed, EMBASE, PsycINFO, Web of Science, and Google Scholar. A total of 3,846 records were retrieved from the database search. Following the removal of duplicates, 3,479 remained for the title/abstract screen and 149 were selected for the full-text review. Of these, 52 were included in the final review and relevant data were extracted. The 52 articles included in the final review consisted of; 11 studied perceptual outcomes only, 16 studied acoustic outcomes only, and 14 studied both perceptual and acoustic outcomes. 13 of these investigated acoustic features that could be used for mask classification. Although the findings varied from article to article, many trends stood out. Many articles revealed that face masks act as a low pass filter, dampening sounds at higher frequencies; however, the frequency range and the degree of attenuation varied based on face mask type. All but five articles that reported on perceptual outcomes showed a common trend that wearing a face mask was associated with poorer speech intelligibility. The findings of the scoping review provided evidence that facial coverings negatively impacted speech intelligibility, which is likely due to a combination of auditory and visual cue degradation. Due to the continued prevalence of mask use, how facial coverings affect a wider variety of speaker populations, such as those with communication impairments, and strategies for overcoming communication challenges should be explored.

**Competing interests:** The authors have declared that no competing interests exist.

## Introduction

The COVID-19 pandemic necessitated several restrictions in the interest of protecting public health, including the wearing of face masks by the general population. Throughout the pandemic, mask mandates and policies have varied from state to state and across the globe [1, 2]. During major waves, the United States Centers for Disease Control and Prevention (CDC) recommendations were for all unvaccinated individuals over the age of two to wear face masks indoors, and for vaccinated individuals to wear face masks in areas of high transmission. Mask recommendations from the CDC include masks that completely cover the nose and mouth and are made of two or more layers of breathable fabric [3]. Early anecdotal reports highlighted the unique challenges to communication brought about by wearing a covering over one's mouth and nose. While face masks had previously been used to protect against disease in, for example, health care settings, never before had their use been so highly encouraged across a vast population.

While the COVID-19 pandemic has resulted in a surge of new research focusing on the effects of face coverings on communication, attempts at characterizing these challenges date back many decades. For example, [4] found that individuals working in toxic environments indicated that they would often remove respirators since talking with respirators was too difficult. With face mask use now ubiquitous, it is important to investigate their impact on everyday communicative settings. A recent literature review investigating the effects of respirators on speech, found that respirators impact speech intelligibility and verbal communication [5]. This literature review related findings back to the burden in medical settings and many reports have been on settings such as professional healthcare or occupational settings in which a face mask was necessary.

Face coverings used to protect the wearer from inhaling unwanted particles likely impact communication due to a combination of acoustic and visual disturbances. Any material that blocks the mouth and nose has the potential to impact the speech signal as well as block or distort visual facial cues for listeners. These characteristics, and consequences, may be shared by other types of face coverings, such as alternative coverings that are now more frequently worn due to the COVID-19 pandemic. Given the widespread use of face coverings in the general public, a better understanding of their impact on communication is needed.

The purpose of this scoping review is to evaluate the state of the evidence at a pivotal point in time (mid-2021, at the height of the COVID-19 pandemic) regarding how face coverings, including those that were recommended for and available to the public and not exclusively health care professionals, impact transmission of the speech signal by characterizing their acoustic and perceptual consequences. In the sections following we discuss the role of masks, standardization of masks and the potential burden of face masks on communication. To the authors' knowledge, this is the most comprehensive review on the topic to date. Given the dynamic nature of this topic, further reviews are likely warranted in the future as more studies are completed. The ongoing information gathering will serve to aid in public policy in the event of future respiratory viruses, and highlight areas in which evidence is lacking. The following sections describe the role of face masks, how their use may be characterized and standardized, and a description of the potential burden they pose on communication.

### Roles and standardization of face coverings

**Medical grade masks and respirators.** The CDC define a respirator as "a personal protective device that is worn on the face or head and covers at least the nose and mouth" [6]. The use of a respirator is recommended not only to prevent the spread of airborne disease, but also to prevent inhalation of hazardous particles that may be transmitted via gases or vapors. The term "respirator" is often used to distinguish a class of face coverings designed to filter out

very small particles in the air. Other types of face masks used in medical settings, such as surgical masks, may not be designed to filter out small particles, but may still protect against larger droplets. The CDC has outlined seven types of respiratory protective devices: filtering facepiece respirators, elastomeric respirators (half and full facepiece), powered air-purifying respirators, supplied-air respirators, self-contained breathing apparatus, and combination respirators [7]. Of these, all but the filtering facepiece respirators (which includes the N95) are reusable and either supply air or make use of a filtration cartridge or canister. These may be used not only in healthcare settings, but in military combat in which toxic fumes are a hazard (e.g., gas masks). Filtering facepiece respirators, conversely, are disposable half-facepiece masks that typically provide protection against particles but not vapors or gases.

Medical grade face coverings will often be described in terms of a standard degree of filtration. Face coverings designed and produced for use in situations where a high degree of filtration is necessary (e.g., healthcare settings) will typically be required to meet certain standards. In the US, for example, face coverings classified as respirators, such as those described above, must meet the standards of the National Institute for Occupational Safety and Health (NIOSH) [8] and may fall into one of several classes that describe the degree of resistance and percentage of filtration of suspended particles [9]. For example, N95 and FFP2 masks are both designed to filter 95% of suspended particles and are "approximately equivalent" [9]. KN95 masks do not necessarily meet United States NIOSH standards but are designed to have the same filtration properties [10]. About 60% of the KN95 respirators in the United States are counterfeit and do not meet NIOSH standards [11]. N95 respirators are the most widely available NIOSH approved respirator [11]. Note that in this review, the term "face mask" is often used broadly to include masks that would also be classified as respirators, such as N95s.

In addition to degree of filtration, respirators may also be categorized as disposable or non-disposable. Disposable respirators consist of N95 and KN95 masks and pattern like face masks. Non-disposable respirators consist of powered air-purifying respirators and elastomeric half facepiece respirators. These respirators have air-purifying components to them and can filter out particles such as dust, and fumes [12].

Surgical masks are a commonly worn medical grade facial covering. Prior to the COVID-19 pandemic, surgical masks were mainly worn in a medical setting by healthcare personnel as a physical barrier to protect both the patient and the healthcare personnel [13]. Surgical masks, which may also be referred to as disposable or medical procedural masks, are not NIOSH approved; however, they are cleared for use by the Food and Drug Administration [13].

Dust respirators are a disposable, non-medical class of respirators that are used to protect against dust during activities in which non-toxic particles may be present, such as mowing the grass or woodworking. Although some dust respirators may resemble N95 respirators, dust respirators are not NIOSH approved. Dust respirators offer a one-way protection only and are not recommended to be used if being exposed to hazardous environments.

**Non-medical grade face coverings used as a preventative measure for disease transmission.** Due to the limited availability of medical grade face coverings at the onset of the COVID-19 pandemic, there was a rise in the recommendation and implementation of non-medical grade masks. An example of these are fabric masks. Fabric masks are now widely available and can be made at home or purchased. Cloth masks may have one layer, multiple layers or multiple layers separated by disposable filters. The level of protection provided from fabric masks are dependent on the layers of fabric and the type of fabric [14]. The CDC recommended wearing cloth masks with multiple layers and a nose wire to ensure proper protection [11]. During the colder months many may wear scarfs, ski masks and balaclavas; however, the CDC does not consider these substitutes as face masks and recommends these be worn over face masks.

A rise in the use of transparent face masks occurred as a means of providing face coverings that provided visual access to the mouth during spoken communication. Many clear face masks and cloth masks with clear window inserts are available on the market today. These types of masks were designed in order to provide access to visual information provided by the talker's mouth when wearing a mask. The rationale here is that visual cues may be especially helpful when communicating with individuals who are hard of hearing or have a disability, young children who may be learning to read or learning a new language, and individuals who need to see the proper shape of the mouth. However, many types of transparent material may provide additional acoustic challenges due to the thicker, potentially reverberant materials used. The FDA recently approved of a transparent medical face mask and indicated that these should be reserved for professionals and patients who require them [11]. Face shields are another type of transparent face covering. Face shields are typically constructed of materials such as polycarbonate or acetate and consist of a rigid, transparent visor that is often open at the bottom and is attached to a frame worn on the head [15]. They are considered adjunctive personal protective equipment and were not recommended to be used in place of, but rather in addition to face masks by the CDC as their effectiveness is not well established and they are not designed to protect against respiratory droplets [12, 15, 16].

**Other types of face coverings.** Many types of face coverings are used for reasons other than to prevent the spread of droplets or particles in the air. Full head enclosures designed to protect against head trauma also often cover the face, such as motorcycle helmets. Activities requiring supplied air rely on full head enclosures as well, such as underwater or space travel. Face coverings have been recommended as a means to protect against poor air quality, such as during fires and highly polluted areas [17, 18].

Face coverings are commonplace in certain religious settings. For example, it is commonplace in certain Muslim communities for women to veil themselves in a niqab or a burqa, both of which cover part of or all of the face, including the mouth and nose. Certain garments designed to protect against the cold are designed to be worn over the face, such as balaclavas (which may or may not include a mouth hole), scarves, and neck gaiters. Disguise, either for entertainment or for criminal activity, may also use face coverings such as masks made of various materials, but may also include other types of covers such as balaclavas.

## Potential burden of masks

While in many cases the purpose of masking is to protect from hazardous conditions, a potential indirect burden of wearing a face covering is the detrimental impact it may have on communication. In a survey conducted of medical personnel in a hospital in Toronto, Ontario during the Severe Acute Respiratory Syndrome (SARS) outbreak, 47% of hospital workers indicated that wearing PPE was associated with communication difficulty [19]. Due to an increase in the use of face masks all over the globe, this same communication difficulty may be a burden to many.

Face masks may pose a burden to communication for potentially three main reasons. First, relevant visual information is lost due to the covering of the mouth, obscuring lip movements and facial gestures and expression. Second, the mask itself acts as a physical barrier between the listener and the speech source and may absorb or attenuate acoustic information. Third, face coverings may introduce physiological restrictions or behavioral adjustments that may have a bearing on speech. For example, certain masks, such as fitted surgical masks like the N95, may restrict jaw movement resulting in limited oral opening and changes to the filtered speech signal. Wearing the mask may itself indirectly lead to changes in how speech is produced by the wearer, either due to conscious or unconscious knowledge of the previously

mentioned barriers. These speech modifications will thus also result in a modified acoustic speech signal. The Institute of Medicine has recommended that there be an increased effort to improve speech intelligibility and reduce communication interference while wearing masks. Although there is a lack of research to support these claims, half-face elastomeric respirators are sometimes marketed with claims of improved communication when manufactured with speaking membranes and/or voice amplifiers [20].

As a physical barrier that covers the mouth, some degree of acoustic attenuation is likely to occur with all types of face masks. Acoustic attenuation in this case refers to the degree of sound energy dampened as a result of this barrier. The degree of attenuation and the frequency components affected are likely dependent on the material of the face mask [21]. When choosing a mask an individual may opt to choose a mask with greater protection; however these masks may be associated with greater reductions of sound transmission [22]. In environments where greater medical protection is necessary such as in a hospital setting, communicating with others may be a challenge.

The impact of facial coverings on communication may be exacerbated in certain populations, such as those susceptible to hearing loss. [23] investigated the impact of facial coverings on populations more susceptible to hearing loss such as older adults. These populations may often rely more on visual cues from the speaker to aid in their understanding of speech; therefore, communicating with someone wearing a face mask may hinder their communication [23]. [24] found that both acoustic and visual cues are fundamental in the listeners ability to recognize and perceive speech and the speaker's content. Individuals who have hearing loss, such as older adults, may have greater difficulty understanding speech in masks which may be attributed to the degraded signal and/or loss of visual cues. Given the widespread use of face masks it is important to consider how face masks pose a potential burden to different populations.

### Rationale & objectives of the current study

The rationale for the current review was to evaluate the state of the evidence regarding the impact of facial coverings on speech transmission. Our primary research questions included the following:

1. What is known from existing literature about the effect of face coverings on speech acoustic outcomes?

2. What is known from existing literature about the effect of face coverings on signal-based speech intelligibility?

While there are likely important visual effects of face coverings on a listeners' ability to understand speech, our review focuses on signal-based outcomes to determine the impact of the coverings on the audio transmission and its impact on listeners.

## Methods

### Protocol

This scoping review followed the framework posited by [25]. [25]'s framework includes five core stages: 1) identifying the research question, 2) identifying relevant studies, 3) study selection, 4) charting the data, and 5) summarizing the results. The authors followed the Preferred Reporting Items for Systematic Reviews and Meta-Analyses (PRISMA) Extension for Scoping Reviews guidelines developed by the Enhancing the Quality and Transparency of Health Research Network [26].

### Eligibility criteria

No filters were employed on the search in order to limit bias.

### Information sources & search

In consultation with a librarian at the University at Buffalo, both authors iteratively developed a set of systematic search terminology. The initial search strategy was constructed with the help of the Yale Medical Subject Heading (MeSH) analyzer for use in PubMed using a combination of title/abstract and subject (MeSH) keywords. The following databases were included, with the search terminology translated for each one: PubMed, Embase, PsycINFO, Web of Science, Google Scholar. The final search was executed on May 10, 2021. Entries were exported into Zotero where duplicates were removed, then exported to the Rayyan software program which was used to manage the screening. The full search strategy for all databases can be found in S1 Appendix. The search strategy approach is summarized as:

- [Terms to identify acoustic and perceptual outcomes] AND

- [Terms to identify relevant face coverings] AND

- [Terms to identify studies related to speech and or voice]

### Selection of sources of evidence

The Rayyan software program was used to manage the screening process [27]. Screening followed two phases: a title/abstract screen and a full text screen. Both phases were carried out independently by both authors in the following way. Following the formation of an initial set of inclusion and exclusion criteria, both authors screened a quasirandom subset of 30 articles in order to establish baseline reliability. Agreement at this stage was 96.7% (29/30). Authors then reviewed the articles independently, meeting after every 1000 articles in order to discuss conflicts. After the first 2000 articles, exclusion criteria were slightly modified to account for unanticipated themes, and previously screened articles were reviewed to ensure their fit in the updated criteria. For example, the exclusion of articles in which face masks were used exclusively for speech measurement (e.g., Phonatory Aerodynamic Systems) or treatment (e.g., Continuous Positive Airway Pressure) was added at this stage. The decision to exclude full helmets and full-face respirators (e.g., gas masks) was also made at this stage in order to maintain a focus on the presence of face masks like those recommended by the CDC. Conflicts were resolved through consensus discussion.

**Screening and eligibility.** Articles passing both levels of screening (title/abstract and full text) met the following criteria: 1) involved the study of at least one physical facial covering, 2) provided quantitative measures of speech production or perception, 3) presented new or original data and 4) were written in English. Exclusion criteria included 1) the study of full head enclosures (e.g., helmets, full or half-face respirators), 2) patents, opinion pieces, or media reports that did not present original data, and 3) occlusions exclusively used as a treatment or measurement tool (e.g., continuous positive airway pressure devices, ventilators, or phonatory aerodynamic systems).

### Data charting process

Both authors completed the data charting process by filling out a customized fillable form and spreadsheet. Information extracted from the articles included 1) article information (title, authors, year, country of research), 2) face covering information (number and type of masks

studied, whether a baseline no-mask condition was included), 3) speaker participant and speech methodological design information (speech stimuli versus non-speech auditory stimuli, live speakers versus pre-recorded speech, number and demographics of speakers if applicable), 4) listener participant and perceptual methodological design information where applicable (number and demographics of listeners, whether live listeners were included, perceptual conditions), 5) perceptual and/or acoustic outcome measures, and 6) main acoustic and perceptual results.

**Data extraction fields.**   Charted data for all articles reporting on acoustic and/or perceptual outcomes appears in S1 Appendix. Explanations of the data extraction fields are reported below. No a-priori codes were established given the range of findings across the articles included in the final text review. Primary categories related to the types of masks, participants, audio stimuli, and experimental conditions were charted first in open-text fields. Major categorical themes were later identified and coded in additional columns which are detailed in the results fields below.

- *Outcome categories*: Articles were coded depending on whether they included outcomes related to **speech acoustics**, **speech perception**, **both acoustics and perception**, or **speaker classification**.

- *Number and type of face masks*

- *Speaker and speech source details*: The number of speakers, speaker gender, and age were charted as they pertained to the study. Sound sources other than live talkers, such as head and torso simulators, were charged. Additional relevant information as it pertained to each study, such as speaker training or profession if applicable, were also included in this section.

- *Listener details*: As with speaker details, the number of listeners, listener gender, age, as well as other pertinent details for the study such as listener training were charted as applicable. If non-human listeners were used, such as with automatic speech recognition systems, this information was included.

- *Speech stimuli and conditions*: Details about speech stimuli, such as whether word, sentences, or non-speech audio like pure tones or noise was charted here. This section also included details of the number and type of experimental conditions including mask conditions and, as applicable, other conditions.

- *Details on main acoustic outcomes*

- *Acoustic attenuation and speech intensity outcomes*: This section included any outcomes related to sound attenuation, transmission loss, overall speech intensity, etc.

- *Other main acoustic outcomes*: Details reported here included, for example, outcomes related to changes in voice-quality acoustic outcomes, segmental acoustic outcomes, etc.

- *Acoustic category tallies*: All articles were coded as 1 or 0 with reference to whether they included outcomes related to attenuation, vocal intensity, voice quality, segmental level changes, or other acoustic results. These charting columns were added after the main data extraction had taken place and themes had been identified.

- *Details on main perceptual outcomes*: This section included main results reported on perceptual speech outcomes.

- *Perceptual category tallies*: All articles were coded as 1 or 0 with reference to whether they included perceptual outcomes related to visual information, noise condition comparisons,

listeners with hearing loss, the use of face shields, the use of transparent masks. These charting columns were added after the main data extraction had taken place and themes had been identified.

- *Details on other important results not reported elsewhere*: Additional results details were included here, as appropriate.

### Synthesis of results

All extracted data entered via the custom form were exported into a spreadsheet for analysis. Following the data extraction procedure, the authors completed a secondary charting process to record the unique types of face masks included across all articles and the range of outcomes that were reported. Mask types were grouped into nine categories related to mask type and representation across the included articles: surgical, KN95, N95, shield, shield + mask combinations, transparent, inclusion of carbon filters, and other. For each mask category, this data synthesis, appearing in S1 Appendix, summarized the 1) minimum and maximum values of acoustic attenuation and/or 2) a summary of the whether the masks were reported to impact speech perception. These results along with descriptions of the other charted variables, are reported in the sections below.

## Results and discussion

A total of 3846 records were retrieved from the database search. Following the removal of duplicates, 3479 remained for the title/abstract screen and 149 were selected for the full-text review. An additional 5 were found during the data extraction and included in the final review. In total 52 were included in the final review and relevant data were extracted. The search procedures are included in Fig 1.

The majority of articles included in the final review (n = 38) were published in 2020 or 2021 and in response to the COVID-19 pandemic, as shown in Fig 2. Fourteen were published prior to 2020. Articles were classified into three main categories corresponding to the outcome data they reported: perceptual outcomes, acoustic outcomes, and acoustic classification.

### Types of face coverings

Face coverings included a range of materials, though the majority of studies investigated one or more of the following: medical grade masks (e.g., surgical masks, N95 respirators), face shields and other types of transparent coverings, and various types of fabric coverings. A summary of the mask types used in the included articles is presented in Fig 3 and Tables 1 and 2, including a summary of the impacts on acoustic attenuation and perceptual findings, as applicable.

### Perceptual outcomes of masks

In total, 21 articles investigated the perceptual outcomes of face coverings. Of these, 12 reported exclusively on perceptual outcomes, including word recognition and sentence intelligibility, and 9 articles reported on both acoustic and perceptual outcomes.

While many types of face masks worn by a speaker were associated with poorer listener accuracy, this was mediated by the presence of visual information, type of material, environmental conditions (e.g., noise, reverberation), the presence of visual cues, and listener characteristics (e.g., hearing loss). Five studies investigated the presence of audio-visual cues from masks on speech perception. Three studies included listeners with hearing loss. Eight studies

**PRISMA 2020 flow diagram for new systematic reviews which included searches of databases and registers only**

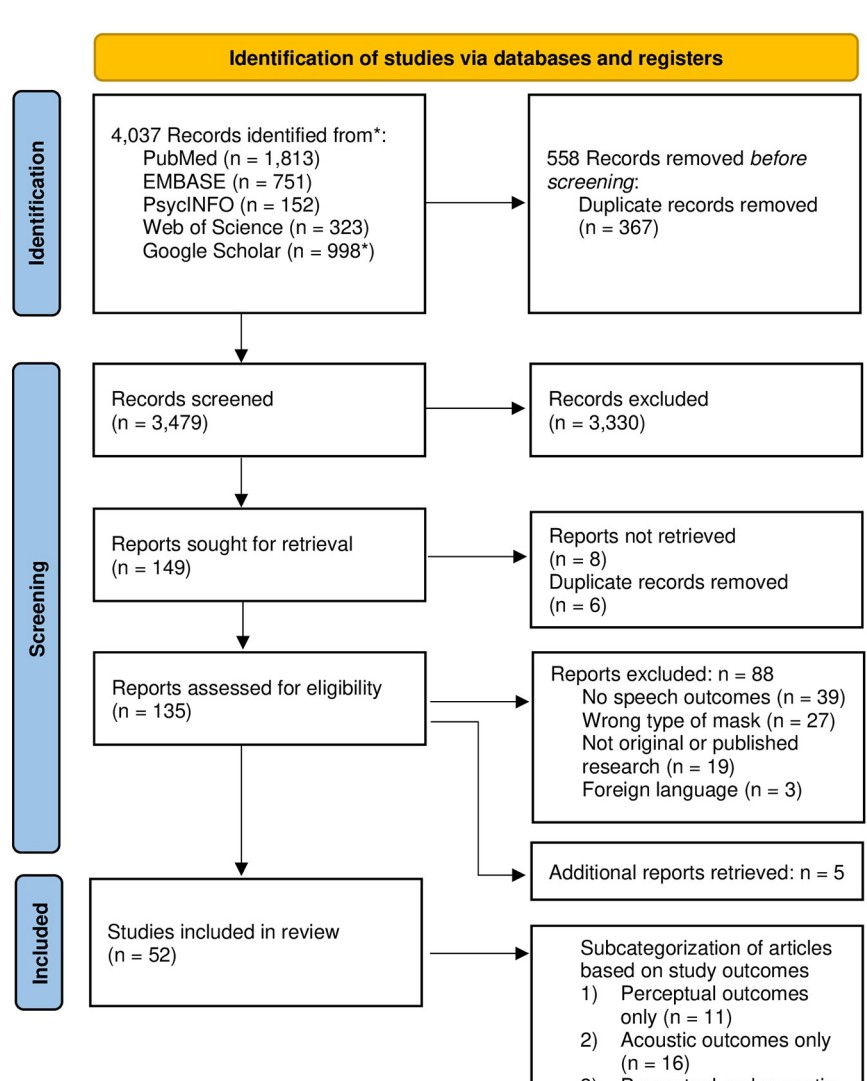

From: Page MJ, McKenzie JE, Bossuyt PM, Boutron I, Hoffmann TC, Mulrow CD, et al. The PRISMA 2020 statement: an updated guideline for reporting systematic reviews. BMJ 2021;372:n71. doi: 10.1136/bmj.n71

For more information, visit: http://www.prisma-statement.org/

**Fig 1. PRISMA flow diagram.**

varied the noise level presented to listeners. Based on themes identified in the articles during the data charting process, the findings of these articles are summarized in the sections below.

**Hearing status of the listener.** Two articles in this review reported on perceptual outcomes of masks for individuals with hearing loss [31, 32]. Overall, results indicate that certain face masks result in greater relative difficulties for those with severe-to-profound hearing loss, though the presence of visual cues aided in speech perception accuracy. [31] reported that the presence of visual cues of the talker afforded by transparent masks to benefit the speech

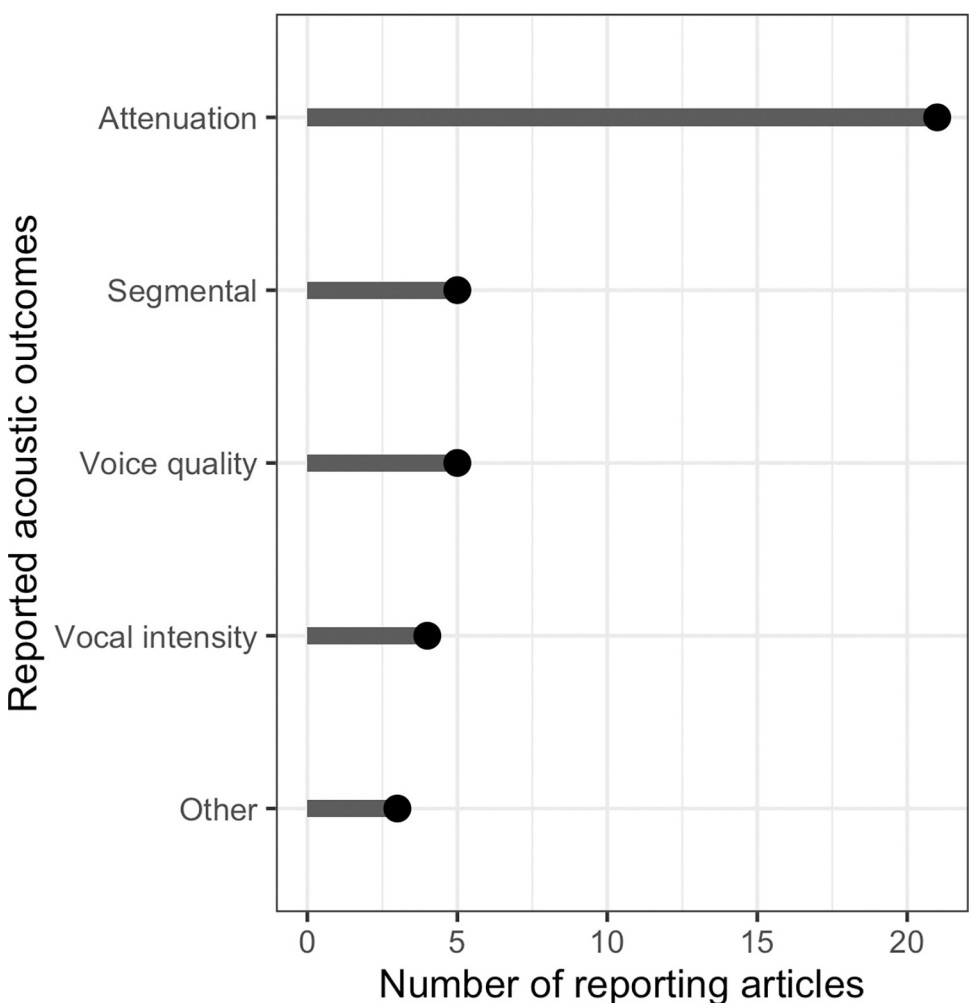

**Fig 2. Timeline of article publication.**

perception abilities of listeners with severe-to-profound hearing loss. This study included ten normal hearing listeners, ten with moderate sensorineural hearing loss, and ten with severe-to-profound hearing loss. Five in the severe-to-profound group used cochlear implants either in combination with hearing aids (n = 3) or alone (n = 2). The authors tested the listeners' speech perception without a mask, with a transparent mask, and with a paper mask, in auditory only and auditory-visual conditions. All speech perception testing was done in 10 dB multi-talker babble and presented at 65 dB HL. The authors found that the presence of visual information was of greater relative benefit to participants with severe-to-profound hearing loss. Listeners with moderate loss performed similarly to those without hearing loss; neither paper nor transparent face masks resulted in significantly worse performance. The severe-to-profound group, on the other hand, benefitted from being able to see the talker's mouth through the transparent mask.

[32] reported on the speech recognition of 23 cochlear implant users when the talker wore an N95, N95 plus shield, or no mask. Speech recognition was measured as percent words correctly repeated from a standardized sentence list (AzBio) tested in quiet listening conditions at 60 dB SPL. Compared to when the talker wore no mask, listeners were significantly less accurate when the talker wore an N95 mask plus a face shield. The N95 by itself was not

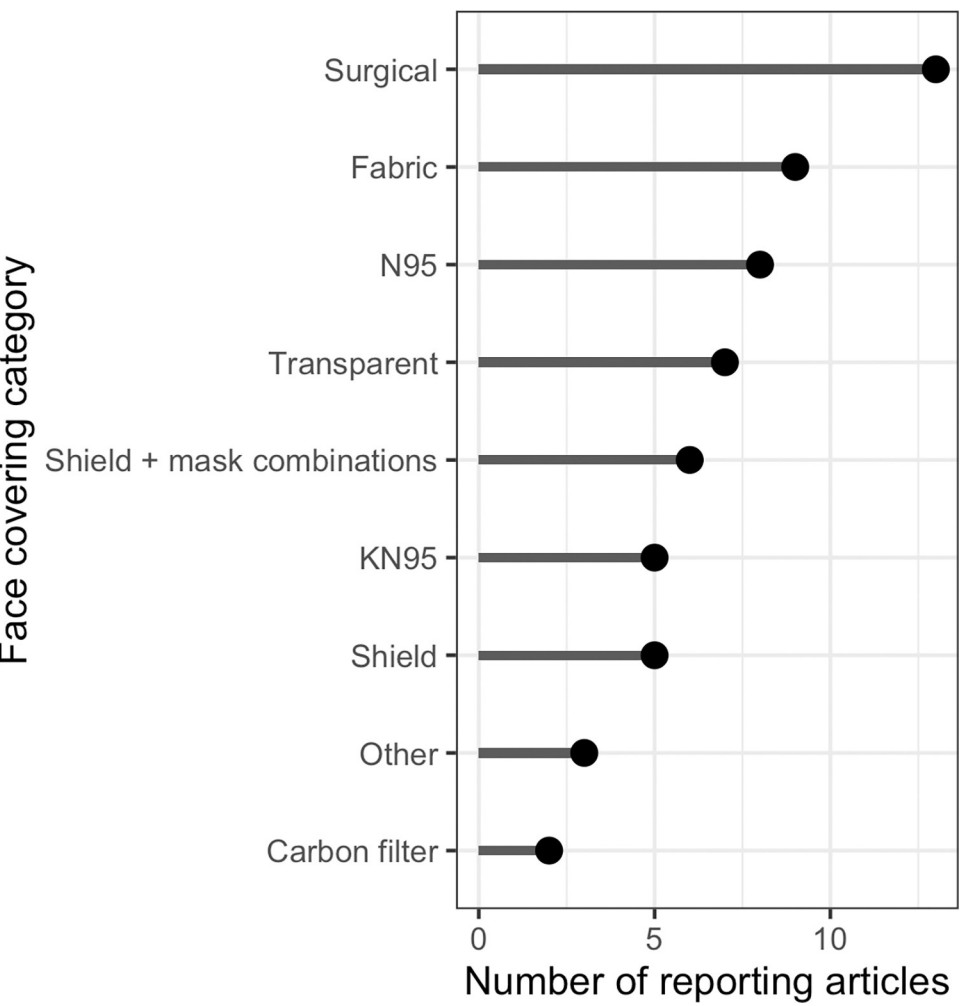

**Fig 3. Types of face coverings.**

significantly worse than baseline. The authors found a correlation between accuracy in the N95 plus shield condition and word recognition in a baseline clinical testing procedure (without a mask). This finding suggests that individuals with lower baseline clinical speech recognition results also struggled more to understand the talker with the N95 and shield.

**Auditory-only versus auditory-visual impacts.**　A subset of studies compared the effect of including or excluding visual information on auditory perception of talkers wearing masks. Overall, the outcomes of these studies support the additional influence of visual cues on auditory-perception of speech when masks are worn [24, 31, 33–35]. Of these, one study included audio recorded without a mask, but included audio-visual scenes in which the talker did or did not wear a mask [33]. In this study, which simulated a video-conference call, listener participants were more accurate in word identification, more confident, and perceived less listener effort when they could see the talkers' mouths than when they heard the *same audio* but saw the talker wearing a surgical mask [33]. These results suggest the role of visual feedback as distinct from acoustic filtration of masks.

The remaining studies comparing audio only and audio-visual conditions included audio recorded when the talkers actually were or were not wearing masks, and presented this audio to listeners with and without accompanying video [24, 31, 34]. These combined results

**Table 1. Range of main perceptual results by mask type.**

| Mask category | Mask type | Number of articles reporting of perceptual outcomes | Perceptual results (compared to baseline) | Summary |
|---|---|---|---|---|
| Surgical | Surgical mask Polypropylene ASTM Level 2 (MediCom 2142) Polypropylene ASTM Level 3 (DemeTECH) | 5 | • lower % accuracy than no mask condition in individuals with moderate hearing loss (Atcherson et al., 2017) • weak but significant effect of modality on consonant identification (subjects identified more consonants during condition when mouth could be seen) (Fecher & Watt, 2013) • masks did not have detrimental impact on speech understanding (Mendel et al., 2008) • mask condition was significant % words correct = 70% (baseline = 80%)(Randazzo et al., 2020) • no effect (Toscano et al., 2021) | Change in perceptual accuracy:—reduction: n = 3- no change: n = 3 |
| Fabric | Fabric mask (various): Variations: with and without filter, single vs. multi-layer; weave/ material | 5 | • no effect on word identification (Cohn et al., 2021) • significant effect of mask on speech perception (Rudge et al., 2020) • significant effect (Toscano et al., 2021) • no significant interaction (Truong et al., 2021) [28] | Change in perceptual accuracy:—reduction: n = 3—no change: n = 3 |
| KN95 | NA | 0 | NA | No studies in this review reported on perceptual outcomes of KN95 masks |
| N95 | N95/FFP2/FFP3 | 2 | • mask condition was significant % words correct = 63% (baseline = 80%) (Randazzo et al., 2020) • significant at low SNR but insignificant at high SNR (Toscano et al., 2021) • not significant (Vos et al., 2021) | Change in perceptual accuracy:—reduction: n = 2—no change: n = 1 |
| Shield | Face shield | 1 | • significant effect of mask on speech perception (Rudge et al., 2020) | Change in perceptual accuracy:—reduction: n = 1—no change: n = 0 |
| Shield + mask combinations | Combination mask + face shield | 3 | • N95 + face shield: speech reception threshold ranged from 15 dB—50 dB; speech discrimination score ranged from 90–95% (Bandaru et al., 2020) • FFP3 + head visor: no significant different in two environments (office & emergency) but showed significant different in the intensive care unit and operating theatre. higher background noise in ICU and operating theatre. (Hampton et al., 2020) • speech recognition was significant for n95 +face shield (Vos et al., 2021) | Change in perceptual accuracy:—reduction: n = 3—no change: n = 0 |
| Transparent | transparent ("see-through") prototype surgical face mask (Atcherson et al., 2017) windowed cloth mask, fully transparent ClearMask (Rudge et al., 2020) custom made transparent mask double-layered cotton + vinyl window (Thibodeau et al., 2021) | 3 | • %Correct: 25 to 100% (Atcherson et al., 2017) * reported on individuals with no HL, mod HL and SEV HL * also had a condition that included audiovisual component • significant effect of mask on speech perception (Rudge et al., 2020) • performance lower when compared to no mask (Thibodeau et al., 2021) * used only transparent mask and for opaque condition put fabric to cover window | Change in perceptual accuracy:—reduction: n = 3—no change: n = 0 |

*(Continued)*

**Table 1.** (Continued)

| Mask category | Mask type | Number of articles reporting of perceptual outcomes | Perceptual results (compared to baseline) | Summary |
|---|---|---|---|---|
| Carbon filter | NA | 0 | NA | No studies in this review reported on perceptual outcomes of Carbon Filter masks |
| Other | a balaclava with a mouth hole, a balaclava without a mouth hole, a motorcycle helmet, a hooded sweatshirt (hoodie) and scarf combination, a niqāb, a rubber mask, and a piece of tape across the talker's mouth. (Fecher & Watt, 2013) | 3 | • weak but significant effect of modality on consonant identification (subjects identified more consonants during condition when mouth could be seen) (Fecher & Watt, 2013) | Change in perceptual accuracy:—reduction: n = 1—no change: n = 0 |

demonstrated that listeners were less accurate in consonant identification in audio-only conditions compared to audio-visual conditions in which talkers' mouths and/or faces were covered [24, 31, 34]. This suggests that acoustically driven challenges in understanding speech with masks may be somewhat alleviated when the talker is seen, even if the mouth is covered. [24] studied audio-only and audio-visual consonant recognition for talkers donning eight distinct types of face concealment. The authors found that, overall, consonant identification accuracy in audio-visual conditions exceeded the audio-only condition, especially in the presence of background noise, and that this effect was stronger for some types of coverings over others. [34] found that while listener accuracy was lower in the audio-visual condition for three types of face coverings (niqab, surgical mask, balaclava), the overall number of consonant misperceptions was very small (2%). These confusions were driven by relatively few error types. The most prominent error types included confusion of stops with fricatives (/t/ ~ /θ/), stop voicing, and place of articulation for stops, fricatives, and nasals. Conversely, [31] found that listeners with typical hearing performed at ceiling regardless of whether the speaker did or did not wear an opaque, paper mask, and access to visual information in the no-mask condition did not alter this. Listeners with hearing loss, however, did perform better in conditions when they could see the speaker's mouth. [36] did not compare with an auditory-visual condition, but reported lowest listener accuracy when speakers wore N95 masks compared to cloth or surgical masks. Surgical and cloth masks also led to degraded accuracy compared to no mask, but incorrect responses were often phonetic approximations, while N95s included a greater relative number of non-responses overall [36].

This last study [31] was one of two that investigated the audio-visual effects of transparent masks on speech perception and included listeners with and without hearing loss [31, 35]. Combined, findings suggest that when the talker is visible, transparent masks which allow the mouth to be seen are associated with increased listener accuracy, but when the talker is not visible, transparent masks are associated with decreased accuracy [31, 35]. Additionally, [35] found listeners reported higher confidence and lower concentration required to understand sentences spoken with a transparent mask compared to an opaque mask when the talker was visible. [31] found that visual cues from transparent masks aided individuals with moderate to severe hearing loss but not those with normal hearing. In particular, listeners with moderate hearing loss performed better when the speaker wore a transparent mask than when they wore an opaque, paper mask, despite of the increased acoustic attenuation found associated with the

**Table 2. Range of main acoustic results by mask type.**

| Mask category | Mask type | Number of articles reporting of acoustic outcomes | Acoustic attenuation (RMS, db SPL) (compared to baseline) | Summary |
|---|---|---|---|---|
| Surgical | Surgical mask Polypropylene ASTM Level 2 (MediCom 2142) Polypropylene ASTM Level 3 (DemeTECH) Polypropylene (YY/T 0969) (Corey et al., 2020) procedure mask (Giuliani et al., 2020) | 8 | • 5 dB reduction (Atcherson et al., 2020) [29] • 3.6 (3ft) - 4.2 dB (6ft) attenuation compared to no mask * 2 to 8 kHz (Atcherson et al., 2021) • 3.6 dB (peak) (Corey et al., 2020) • 3–4 dB reduction over 1.5 kHz (Giuliani et al., 2020) • 2 dB reduction (Nguyen et al., 2021) [30] • 6 dB reduction (Pörschmann et al., 2020) ** estimate based on figure • no attenuation (Wolfe et al., 2020) • 14 dB (Saeidi et al.,2016) | Range of acoustic attenuation: ~0 dB (Wolfe et al., 2020) to 14 dB (Saeidi et al., 2016). |
| Fabric | Fabric mask (various): Variations: with and without filter, single vs. multi-layer; weave/material Cotton jersey (generic) (Corey et al., 2020) Cotton plain (handmade) Cotton/spandex jersey (generic) Cotton/spandex jersey (LASC) (Corey et al., 2020) Cotton plain and denim (Jo-Ann)(Corey et al., 2020) Cotton percale bedsheet and polyester trim (handmade)(Corey et al., 2020) | 4 | • 5.1 dB (6ft handmade fabric) - 6.1 dB (3ft handmade fabric with HEPA filter) attenuation compared to no mask * 2 to 8 kHz (Atcherson et al., 2021) • 4 dB—12.6 dB (peak) (Corey et al., 2020) • 14 dB reduction (Pörschmann et al., 2020) ** estimate based on figure • 2–3 dB of attenuation from 4,000 to 8,000 Hz (Wolfe et al., 2020) | Range of acoustic attenuation: ~0 dB (Wolfe et al., 2020) to 14 dB (Saeidi et al., 2016). |
| KN95 | KN95 | 5 | • 8.7 dB reduction (Atcherson et al., 2020) [29] • 6.3 dB attenuation compared to no mask * 2 to 8 kHz *distance (3ft vs 6ft) made no difference * 2 to 8 kHz (Atcherson et al., 2021) • 4 dB (peak) (Corey et al., 2020) • 5.2 dB (Nguyen et al., 2021)[30] • 8 dB reduction (Pörschmann et al., 2020) ** estimate based on figure | Range of acoustic attenuation: 4 dB (Corey et al., 2020) to 8.7 dB (Atcherson et al., 2020). |
| N95 | N95/FFP2/FFP3 | 7 | • 10.9 dB reduction (Atcherson et al., 2020) [29] • 6.2 dB (6ft) - 6.4 dB (3ft) attenuation compared to no mask * 2 to 8 kHz (Atcherson et al., 2021) • 5.7 dB (peak) (Corey et al., 2020) • Reduced intensity by 3–10 dB over 1kHz (Giuliani et al.,2020) • 10 dB (Randazzo et al., 2020) • 4.3 db (Vos et al., 2021) • 2–3 dB of attenuation from 4,000 to 8,000 Hz (Wolfe et al., 2020) | Range of acoustic attenuation: 2 dB (Wolfe et al., 2020) to 10.9 dB (Atcherson et al., 2020). |
| Shield | Face shield plastic shield (generic) (Atcherson et al., 2021) Humanity shield (Rapid Response PPE) (Atcherson 2021) Moog plastic shield with apron (handmade) (Atcherson et al., 2021) | 4 | • 13.7 (6ft) - 17.2 (3ft) attenuation compared to no mask * 2 to 8 kHz (Atcherson et al., 2021) • face shields "split" sound propagation, mostly at higher frequencies (caniato2021) • 13.7 dB (peak) (Corey et al., 2020) • 3 to 6 dB of attenuation from 4,000 to 8,000 Hz (Wolfe et al., 2020) | Range of acoustic attenuation: 3 dB (Wolfe et al., 2020) to 17.2 dB (Atcherson et al., 2021). |

*(Continued)*

**Table 2.** (Continued)

| Mask category | Mask type | Number of articles reporting of acoustic outcomes | Acoustic attenuation (RMS, db SPL) (compared to baseline) | Summary |
|---|---|---|---|---|
| Shield + mask combinations | Combination mask + face shield | 5 | • 20 dB (sm + face shield) - 29.2 db (kn95 + face shield and transparent + face shield) dB reduction (Atcherson et al., 2020) [29]<br>• 18.0 (Polypropylene ASTM Level 3 (DemeTECH) + face shield) - 25.7 (Polypropylene ASTM Level 2 (MediCom 2142) + N95 + face shield) attenuation compared to no mask * 2 to 8 kHz (Atcherson et al., 2021)<br>• face shield + procedure mask: 6–10 reduction over 1 kHz (Giuliani et al.,2020)<br>• FFP3-vent+shield: 2.5–4kHz range: up to 18.3 dB attenuation (Muzzi et al., 2021)<br>• FFP2-ventilated+shield: 12.5–2kHz range: up to 9.7 dB attenuation (Muzzi et al., 2021)<br>n95 + shield: 17.3 db (Vos et al., 2021) | Range of acoustic attenuation: 2.5 dB (Muzzi et al., 2021) to 29.2 dB (Atcherson et al., 2020). |
| Transparent | Transparent face masks:<br>transparent ("see-through") prototype surgical face mask (Atcherson et al., 2017)<br>prototype FaceView transparent mask (Atcherson et al., 2020) [29]<br>Safe 'N' Clear transparent surgical mask (Atcherson et al., 2020) [29]<br>handmade transparent cloth mask(Atcherson et al., 2020) [29]<br>The communicator (Safe 'N' Clear) (Atcherson et al., 2021)<br>Cotton/polyester blend and vinyl window 1 (handmade) (Atcherson et al., 2021)<br>Cotton/polyester blend and vinyl window 2 (handmade) (Atcherson et al., 2021)<br>ClearMask (ClearMask LLC) (Atcherson et al., 2021)<br>Cloth and vinyl window (handmade) (Corey et al., 2020)<br>Cloth and PVC window (UTSDesignStore) (Corey et al., 2020) | 6 | • transparent facemasks attenuated more than non-transparent face masks (Atcherson et al., 2021; Atcherson et al., 2017)<br>• 8.5 (6ft)–16.1 (3ft) attenuation compared to no mask * 2 to 8 kHz (Atcherson et al., 2021)<br>• 10.8–12.5 dB (peak) (Corey et al., 2020)<br>• 11.32 dB attenuation for transparent, 13.64 dB for opaque (Thibodeau et al., 2021) * used only transparent mask and for opaque condition put fabric to cover window<br>• Communicator(TM) transparent mask: 8.4 dB<br>• ClearMask(TM): 10.9 dB (Vos et al., 2021) | Range of acoustic attenuation: 8.4 dB (Vos et al., 2021) to 16.1 dB (Atcherson et al., 2021). |
| Carbon filter | carbon filter mask (PM2.5), | 2 | • 8.0 (3ft)–8.4 dB (6ft) attenuation compared to no mask (Atcherson et al., 2021)<br>• 6 dB reduction (Pörschmann et al., 2020) ** estimate based on figure | Range of acoustic attenuation: 8 dB to 8.4 dB (Atcherson et al., 2021). |
| Other | Motorcycle helmet, latex rubber mask, scarf, balaclavia, plastic party mask | 2 | • scarf: 2.5 dB @ 4.1 kHz (Saeidi et al., 2016)<br>• rubber mask: 30 dB at 5.2 kHz. (Saeidi et al., 2016) | Range of acoustic attenuation: 2.5 dB (Saeidi et al., 2016) to 30 dB (Saeidi et al., 2016). |

transparent mask. This finding highlights the potential importance of visual cues especially for those with hearing loss.

Relatedly, six studies examined the effect of layering a face shield over a mask on speech perception in an auditory-only context [32, 37–41]. In all cases, layering a face shield on top of a mask was associated with reduced intelligibility. These findings suggest the addition of a face shield disrupts acoustic transmission of the speech signal resulting in poorer listener accuracy. The increased perceptual difficulties in auditory-only scenarios when a transparent mask or shield is used are likely related to increased acoustic attenuation and/or distortion from the rigid materials [42].

**Varying levels of noise.** Eight studies explored the effects of different noise levels on intelligibility of speech in masks [24, 34, 37, 40, 43–46]. Overall, results suggest intelligibility deficits of masks are worse in the presence of higher noise levels or in environments with greater reverberation. Strategies for improving speech intelligibility in masks included speaking more loudly [40] or more clearly [47] or using amplification such as an in-ear listening device [48] or a microphone [41].

**Minimal impact.** Although a common trend across all articles was that wearing a mask was typically associated with poorer speech intelligibility compared to not wearing a mask, this was not the case across the board. Six articles reported minimal to no effects of masks on speech perception, most often for surgical masks [32, 34, 38, 43, 44, 47]. [44] found no difference in intelligibility of speech produced with a surgical mask compared to without, regardless of the presence or absence of background noise. [32] found no differences in listener accuracy (evaluated in quiet) when talkers donned an N95 mask compared to without, but did find lower accuracy when the N95 mask was layered with a face shield. Similarly, [38] reported descriptive statistics on preliminary data suggesting that decreases in accuracy (in the presence of multitalker background noise) were observed when talkers wore face masks and further when layering with a face shield, they noted that the relatively modest changes would likely not yield statistically meaningful differences. The authors noted a ceiling effect for the unmasked condition, indicating that the listening conditions may not have been challenging enough. [34] found that speech with surgical masks, compared to niqabs and balaclavas, were the easiest to understand and were also associated with less acoustic transmission loss. The perceptual task was completed in quiet listening conditions. While [43] found that listening effort did not differ significantly as a function of mask type, the authors reported that surgical masks had lowest probability of experiencing greater listening effort followed by fabric and the greatest listening effort was for N95s. The authors found this pattern consistent in two acoustic room simulations designed to replicate both low and high reverberation conditions, consistent with easier and harder listening conditions. [47] found no effect of a fabric face mask conditions when talkers spoke in a habitual speech style (in -6 dB SNR of multitalker babble), but speech produced with a mask was *more* intelligible than without a mask when talkers were instructed to speak clearly. The authors attributed this to a targeted adaptation approach in which talkers overcompensated for the mask when adapting a clear speech style for listener comprehension.

While not consistently controlled across all studies, the degree of additional listening challenges, such as the presence of background noise or hearing loss, appears to be a likely contributing factor. More challenging listening conditions may be more sensitive to subtle perceptual differences with masks. On the other hand, masks, and in particular surgical masks, may not pose significant listening barriers in more favorable listening conditions and/or for listeners who have typical hearing thresholds.

## Acoustic outcomes

Of the 52 total included articles, 28 investigated the effect of face coverings on speech acoustic outcomes, which in most cases characterized the degree of attenuation of the speech signal. Other acoustic outcomes included overall speech intensity (n = 4), voice quality-related acoustic measures (n = 5), and segmental speech acoustics, such as properties of vowel and fricative productions (n = 5). Key differences in methodological approaches of note were whether the sound source included a live human talker, pre-recorded human talker, or non-speech sound, such as frequency sweeps. The following sections include mention of the range of acoustic characteristics reported in the included articles; the intention of this is to provide an overview rather than a consensus, and the reader is cautioned to recall the extensive methodological

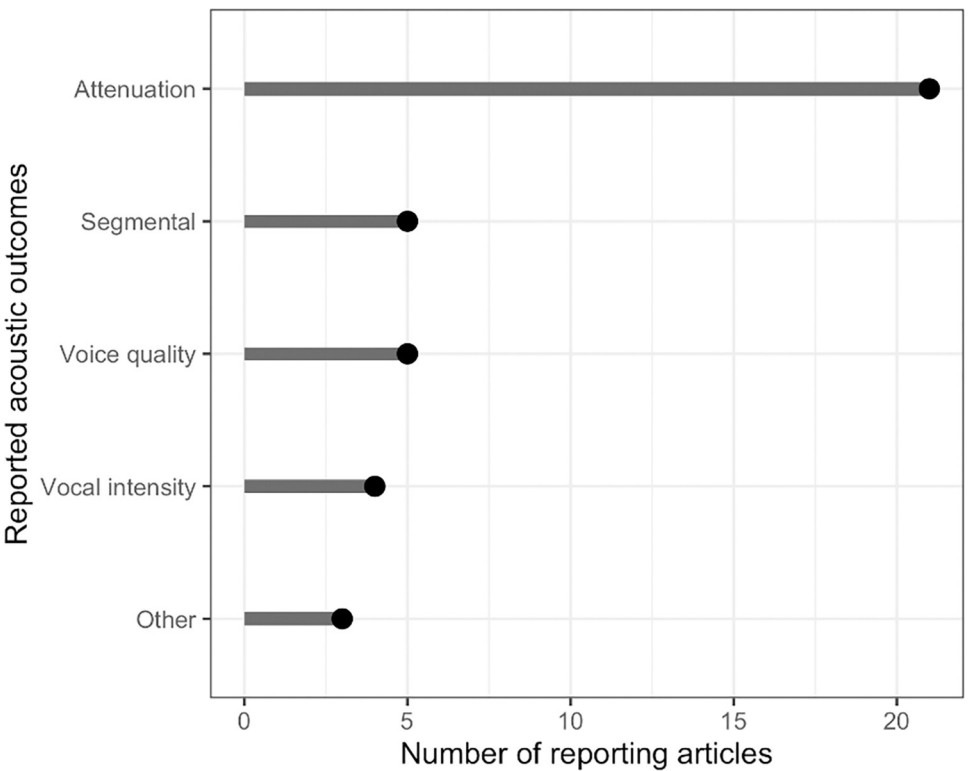

**Fig 4. Number of articles reporting on acoustic outcomes.**

variability in the acoustic results presented below. The distribution of articles reporting on acoustic outcomes is shown in Fig 4.

**Acoustic attenuation.**    In total, 21 articles reported on acoustic attenuation imposed by face coverings. Overall, most articles reporting on acoustic attenuation provided evidence that masks act as a low pass filter by dampening the amplitude of frequency components above 1 to 2 kHz. The range of attenuated frequencies and the degree to which masks dampened sound varied across mask types. A summary of the range of acoustic attenuation findings for each mask type is included in Table 1. Studies that compared multiple masks were consistent in reporting that surgical masks demonstrated the least amount of attenuation overall [21, 29, 30, 36, 42, 49, 50]. Attenuation imposed by surgical masks ranged from 0–2 dB in mean spectral amplitude between 1–8 kHz [30, 51] to 14 dB at 4.5 kHz [52]. The greatest attenuation was observed for transparent masks and face shields, ranging from ~3 dB [51] to up to 29.2 dB [29]. Mask material and weave were reported to be the most important variables when considering the degree of attenuation [21]. Surgical masks, some fabric masks, and KN95 masks demonstrated peak attenuation at approximately 4 dB [21, 36, 53], while N95 respirators peaked at approximately 6–10 dB [21, 36, 53]. One study reported two apparent peaks for an N95 worn by a live talker at approximately 10 dB; one at 800 Hz and another at 2–3 kHz [36]. [54] recorded white noise produced via a mannequin loudspeaker positioned six feet from a microphone in four mask and four mask plus face shield conditions. They reported preliminary data, identifying reductions in maximum sound pressure level across a spectral range of 0 to 8kHz. Reductions ranged from 5 dB SPL (surgical mask only) to 29.2 dB SPL (KN95 mask plus face shield and transparent mask plus face shield). [42] expanded on the preliminary findings from [54], including 15 mask conditions (grouped by non-transparent and transparent

types) recorded at a 3 foot and 6 foot distance. They found minimal attenuation below 1kHz, and a wide variation of reductions in the 2–8 kHz frequency range, and greater attenuation at the farther distance. Between 2 and 8 kHz, compared to no mask, surgical masks were associated with the least attenuation (~4 dB RMS at both 3 and 6-foot distances), and the Humanity Shield transparent mask was associated with the greatest attenuation (~17 dB RMS at both distances). Adding a face shield resulted in an additional 10 to 16 dB RMS attenuation, resulting in a total attenuation of 18 to 25 dB RMS. [44] reported on attenuation from one talker producing sentences with and without a standard surgical mask at a 1-foot distance under controlled speaking conditions (i.e., intentionally maintaining a steady duration and intensity). The authors found a small but significant difference in average RMS power across the speech spectra, suggesting that the talkers speech was actually less attenuated with the mask on, on the order of 0.21 dB SPL.

Two articles reported on spectral slope or tilt, which may be considered an indirect measures of attenuation as it captures the difference in low and high energy bands. [30] found no changes in spectral slopes (comparing 0–1 kHz to 1–8 kHz bands) in sustained vowels, but found higher relative low energy values in the surgical and KN95 masks, consistent with previous accounts of low-pass filtering. [55] reported an impact of surgical masks and N95 masks but not fabric masks on spectral tilt. These differences were observed when measured via a headset and tabletop microphone.

**Other acoustic consequences of face coverings.** *Overall vocal intensity-related outcomes.* While higher frequency components of the signal were found to be attenuated by most masks, overall changes in vocal intensity (i.e., the overall acoustic energy in the speech signal) were not consistently found. This finding is not unexpected, given the low-pass filtering effect of face masks and that most energy in speech is concentrated in lower-frequency ranges. Speakers could, however, alter their own behavior when wearing masks, such as by increasing vocal effort, which could lead to increased speech intensity. Of the studies reporting acoustic outcomes, four reported on overall speech intensity [30, 55–57].

Of these, one controlled for behavioral adjustments in response to mask wearing by taking acoustic measurements from prerecorded samples. [56] presented prerecorded speech samples from a customized head and torso simulator loudspeaker with a mouth-to-microphone distance of 8 cm. The authors compared three mask types (surgical mask, FFP2, and transparent mask) and found that while the surgical mask was not associated with differences in speech intensity (extracted from a sustained vowel), the FFP2 and transparent masks did result in speech signals that were approximately 1.5 dB SPL lower compared to no mask. Two studies reported no significant differences in vocal intensity for any of the face masks studied, including surgical masks, KN95 masks, N95 masks, and cloth masks [55, 57]. Stimuli from these studies included sustained vowel production and/or connected speech tasks produced by live talkers. While [57] found no group differences in intensity, they reported that the majority (65% of 60 subjects) produced decreased speech intensity from sustained vowels on the order of 1 to 2 dB with the surgical mask on, while 35% produced an increased intensity. This suggests individual differences in behavioral responses to wearing masks.

While not included in the counts above, two studies reported opposing findings on changes in speech intensity featuring a very small number of trained talkers (one or two) whose speech was used for perceptual testing [40, 47]. [40] measured the speech-to-noise ratio (SNR) of a single trained researcher reading test sentence material in varying noise levels with and without wearing a face covering. In this study, the face covering was a fit-tested FFP3 mask plus a visor, which was included as part of standard hospital PPE. In low levels of background noise (i.e., 45 dB, simulating standard office noise), the talker increased their SNR by ~2 dB more when wearing PPE than without it. This trend generally persisted as noise levels increased,

with the exception of the highest noise levels (70 dB, simulating a surgical operating theatre), at which point SNR with and without the face coverings were approximately the same. The authors did not report any statistical findings for this outcome. [47] reported on speech produced by two talkers producing sentences in three different styles (habitual, emotional, and clear) with and without a fabric mask. While speech intensity was predictably altered across speech styles, the authors found no obvious differences in speech intensity across the mask conditions. It should be noted that acoustic outcomes in [47], including speech intensity, were reported as summary statistics and were not a primary outcome. This study was included in this section, however, to provide a holistic picture of the effects of masks on speech intensity reported so far in the literature.

Overall, these findings suggest that while vocal intensity differences may exist for heavier, thicker mask materials, these differences are small and inconsistent, and may sometimes be attributable to behavioral adjustments to masks. For example, talkers may alter their own speech in response to wearing a mask, which could result in increased intensity. These individual differences could explain the conflicting pattern of results in [56]. Additional claims related to behavioral differences would likely require more studies with a larger number of talkers in order to draw valid generalizations.

*Voice-quality related acoustic outcomes.* Five articles reported on voice and voice-quality-related characteristics, suggesting inconsistent patterns regarding the impact of masks. Four of these reported on live talkers [30, 55, 57, 58] and one reported on pre-recorded speech samples [56]. Overall, face masks (including surgical masks, FFP2 masks, and transparent masks) were not found to impact fundamental frequency [55–58].

The impact of masks on voice quality characteristics such as harmonics-to-noise ratio (HNR) and cepstral peak prominence smoothed (CPPS) varied substantially across the five studies. Three of the five articles found no differences in HNR or CPPS in unmasked versus surgical mask conditions in live talkers [55–58]. [56], reporting on prerecorded sustained vowels, found that while surgical and FFP2 masks were not associated with significant differences in HNR compared to no mask, transparent masks were. All three mask types were associated with decreases in CPPS. [30] found that wearing either a surgical or KN95 mask was associated with an *increase* in HNR in live talkers producing a sustained /a/ vowel, but neither mask was associated with a change in CPPS (measured in vowel production, sentence reading, and passage reading). The authors suggested these results may be due to talkers subconsciously adopting their phonation style when wearing a mask.

Few differences were reported in jitter and shimmer. Specifically, surgical masks and FFP2 masks were not found to be associated with changes to jitter or shimmer [55–58], while transparent masks were associated with an increase in shimmer but no change in jitter [56]. [56] also reported an increase in AVQI during pre-recorded read sentences across all mask conditions they included (surgical, FFP2, transparent). [55] reported no change in jitter and shimmer for N95, cloth, nor surgical masks.

*Segmental acoustic outcomes.* Five articles reported on segmental acoustic outcomes. Of these, four reported on the speech of live talkers [59–62] and one reported on pre-recorded speech samples [56]. Two articles reported on the effects of face coverings on vowel formant measures [56, 61], and three reported on fricative characteristics [59, 60, 62].

[56] reported on first and second formants and formant bandwidths of sustained /a/ vowels in prerecorded speech played via a mannequin loudspeaker wearing three different masks. An increase in first formant frequencies was found for a transparent mask (113 Hz), while no differences were found for surgical or FFP2 masks. There were no differences in first formant bandwidths for any of the masks. A decrease in second formant frequencies was found for both the FFP2 (56 Hz) and transparent mask (113 Hz), but not the surgical mask, and

differences in second formant bandwidth was found for the transparent mask only. Findings suggest that transparent masks may amplify low frequency characteristics, while masks made of thicker or harder materials may dampen higher frequency characteristics. [61] reported on ten talkers' production of central vowels in Pahari in three face cover conditions: niqab, helmet, and mask (not specified). The authors reported on effects, but not effect directions for F1, F2, and duration of two central vowels. The authors reported an effect of all three face coverings on first and second formants of /ə/ but not /a:/, and inconsistent effects on vowel durations. The authors did not report the direction of effects on these outcomes, thus it is not possible to draw conclusions from this study on the precise nature of the impact of face coverings on vowel production.

The three articles reporting on fricative characteristics measured the impact of face coverings that are often used in order to conceal the face (including, for example, helmets, balaclavas, and party masks in addition to surgical masks). Overall, findings suggest that face coverings of these sorts are associated with lower spectral center of gravity, and inconsistent changes in fricative intensity and skewness and kurtosis (58, 59, 61). Findings overall suggested greater sound absorption for rubber masks, helmets, and tape covering the mouth compared to surgical masks and cloth coverings (hoodies, balaclavas, and niqabs), leading to a shift in spectral characteristics of fricatives. These spectral changes may have implications for consonant identification. It is worth noting that [34], who is not included in the counts here, also made qualitative statements about acoustic characteristics of consonants produced with niqabs, balaclavas, and surgical masks in order to speculate about reasons for listener consonant misidentification.

*Other acoustic outcomes*. Additional acoustic outcomes not reported above were investigated in three articles. These included temporal speech pause characteristics [55] and the acoustic Speech Transmission Index [22, 63]. While mean pause length and variability of pause length did not differ in surgical, N95, and cloth masks compared to baseline, N95s and cloth masks were associated with a higher percentage of pauses.

Two articles reported on the STI [22, 63]. The STI is a predictive measure of speech intelligibility that is calculated based on acoustic signal-to-noise ratios in the presence of pink ambient noise. Findings support previous conclusions of the relatively minimal effect of surgical masks, which were found to yield the smallest relative effect on the STI outcomes [22, 63]. [22] measured the effects of 11 face coverings on sound propagation in high and low reverberant classroom settings using a logarithmic sine sweep emitted from a loudspeaker. Outcomes included including reverberation time, acoustic clarity, and the STI. The authors found that most face masks studied impacted sound propagation, and these differences were increased in rooms with greater reverberation and for male talkers. Surgical masks were least impactful overall. In contrast to previously reported findings of the increased attenuation imposed by transparent masks [21, 31, 32, 35, 42], [22] found that masks with transparent windows did not significantly impact speech transmission, though face shields had the effect of splitting sound propagation directivity in two (above and below the shield). [63] investigated the effect of multiple models of three types of face coverings on the STI (surgical masks, N95 masks, and air-purifying respirators; note that air-purifying respirators were excluded from this review and are therefore not discussed in this section). Surgical masks were associated with the least impact on the STI, followed by the N95 and then by the air-purifying respirators. Specifically, the surgical masks deviated from the no-mask condition by 3–4%, while the N95 masks deviated by 13–17%. Additionally, [47] reported summary statistics of six acoustic measures produced by two talkers with and without a fabric mask across three speech styles. This article is not included in the counts here because they did not include these measures as primary outcomes. The authors reported that no obvious differences in the mask conditions were observed

for any of the measures, which included speech intensity (discussed in the previous section), speech rate, mean f0 and f0 variation, and vowel dispersion.

## Acoustic classification of masks

In total, 13 articles were categorized as reporting outcomes related to acoustic classification of face masks, in which the aim of the authors was to identify whether a speaker was wearing a face mask from a set of acoustic features or algorithms. Of these, seven articles were published in the Proceedings of Interspeech 2020 as a part of the Computational Paralinguistics Mask Sub-Challenge [64–70]. The goal of this challenge was for authors to identify whether a speaker was wearing a surgical mask or not, by using machine learning techniques to identify acoustic features from audio recordings from the Mask Augsburg Speech Corpus (MASC) [71]. The MASC includes recordings of 32 speakers of German engaging in a variety of speech tasks with and without a surgical mask. Speech tasks included passage reading, question responses, reading and repeating words, and spontaneous picture descriptions. Of these, [67] was awarded as the challenge winner for achieving the highest accuracy (80.1% "Unweighted Average Recall") using spectrogram image classification techniques. A summary of techniques used in this challenge are reported in [72].

Of the remaining six articles, three also reported on acoustic classification of the MASC data, but were published in conference proceedings other than Interspeech 2020 [73–75]. One article, published in Interspeech in 2015, reported on a corpus of 8 talkers engaging in sentence reading and spontaneous picture description tasks while wearing one of four types of face coverings: motorcycle helmet, rubber mask, hood and scarf, and a surgical mask [76]. Unlike the other classification articles in this category, the aim of this study was to identify whether an automatic speech recognition system could correctly identify individual talkers across the different face covering conditions.

An in depth summary of the outcomes across these investigations into machine learning algorithms aimed at the identification and prediction of the presence of face coverings is beyond the scope of the present review. Findings suggest, however, that the use and refinement of such algorithms may aid in future descriptions of the acoustic impacts of masks and the ability to monitor the speech of talkers while wearing masks. For a more in depth summary of issues of classification of masks, specifically in the context of the Interspeech 2020 ComParE Mask Sub-Challenge, readers are directed to [72].

## Study limitations

This scoping review provides, to the authors' knowledge, the most comprehensive review of the effects of face masks on spoken communication. This review captures the state of the evidence in mid-2021, at the height of the COVID-19 pandemic. Due to the dynamic, evolving nature of this topic, future reviews are warranted to continue to characterize what is known about the effects of face masks and other forms of personal protective equipment on speech, as well as to identify ongoing gaps. This review sought to characterize the effects of masks recommended for public use by the CDC during the COVID-19 pandemic. As such, this review excluded certain types of face coverings such as commercial, non-disposable respirators. For a review of the effects of respirators on speech intelligibility, the reader is directed to [5]. Given the rapid rise in interest in this topic, the authors chose not to restrict the review to exclusively peer-reviewed academic journals; white papers, magazine editorials, and proceedings papers have been included. Furthermore, as a part of the scoping review approach, no attempts were made to evaluate the quality of the articles included in the review. As such, some of the conclusions of the studies included may report on preliminary data or experimental designs in less

rigorous settings. Similarly, further analysis and synthesis of methodological differences across studies would be warranted in future reviews. For example, the differences across studies that use live talkers versus mannequins, quiet or noisy listening conditions, and the range of recording equipment used would likely provide help to further identify the nature of the effects of face masks and, importantly, methods for remediation. The full results of the data extraction, including citations and specific methodological details of interest are included in the Supplemental Materials to aid the reader in drawing their own conclusions. Another limitation is related to the timely nature of the topic. At the time of writing, a secondary search revealed several new potentially relevant articles. An update to the present review will be warranted, as new research related to the effects of face masks is a topic that is only continuing to grow. The purpose of the current review was to evaluate the state of the evidence at the current point in time.

## Clinical and research implications

Overall, the evidence collected in this review suggest that face masks have been consistently shown to dampen higher frequency information of the speech signal. The effect of face masks on speech perception is less consistent, but studies overall point to a detrimental effect that is worsened by factors such as increased noise and the absence of visual cues. Surgical masks appear to have the least impact on acoustic attenuation and perceptual outcomes. Most soft masks, such as disposable medical grade masks and fabric masks, dampen the acoustic signal to a lesser degree than transparent masks [22]. The thickness and weave of the face covering material appears to play a role in the acoustic impact of the masks [21], while rigid materials such as those used in transparent masks and face shields lead to even greater degrees of attenuation. Transparent masks and especially face shields with visors, though, may also amplify certain higher resonant frequencies and lead to changes in the direction of acoustic transmission [22, 54]. From an auditory perspective, greater dampening appears to be associated with worse speech perception, but the presence of visual cues afforded by clear masks alleviates this effect in some circumstances. Areas of future research would benefit from identifying strategies that aid in overcoming the perceptual challenges of masks, including more behavioral modifications, such as clear speech [47] or loud speech [40] and the use of amplification [48, 41].

A noticeable gap in the literature was related to representative sample populations that would carry clinical implications for the field of speech-language pathology and audiology. While a small number of studies (n = 2) examined the effects of masks on listeners with hearing loss, none reported on other clinical populations such as those with speech or voice disorders. Given the continued prevalence of mask use and the documented acoustic impact of masks, this is an area that should be addressed in the future.

## Summary and conclusion

This scoping review evaluated the state of the evidence regarding the impact of facial coverings on speech transmission at an acoustic and perceptual level. Fifty-two articles were included in the final review, encompassing a wide range of methodologies. Results suggest that face masks consistently are reported to attenuate higher frequency spectral information in the speech signal above 1 to 2 kHz, though this range was found to vary by mask type. Across articles included in this review, face masks are less consistently reported to impact other acoustic features of speech including vocal intensity, voice-quality related measures, and acoustic-phonetic aspects of speech production. While a common trend among the articles reporting on listener perceptual consequences of talkers wearing face masks suggested poorer overall speech intelligibility. However, the presence and magnitude of this effect varied widely and was subject to

changes in background noise levels and listener characteristics. Areas of future work should include a wider range of talker characteristics.

## Supporting information

**S1 Appendix. Full search terminology for scoping review.**
(DOCX)

**S1 Data.**
(XLSX)

## Author Contributions

**Conceptualization:** Gursharan Badh, Thea Knowles.

**Data curation:** Gursharan Badh, Thea Knowles.

**Formal analysis:** Gursharan Badh, Thea Knowles.

**Investigation:** Gursharan Badh, Thea Knowles.

**Methodology:** Gursharan Badh, Thea Knowles.

**Project administration:** Thea Knowles.

**Resources:** Thea Knowles.

**Supervision:** Thea Knowles.

**Writing – original draft:** Gursharan Badh, Thea Knowles.

**Writing – review & editing:** Gursharan Badh, Thea Knowles.

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
