## [Decision Letter · Decision Letter 0]

9 Nov 2022

PONE-D-22-09426Acoustic and perceptual impact of face masks on speech: A scoping reviewPLOS ONE

Dear Dr. Knowles,

Thank you for submitting your manuscript to PLOS ONE. After careful consideration, we feel that it has merit but does not fully meet PLOS ONE’s publication criteria as it currently stands. Therefore, we invite you to submit a revised version of the manuscript that addresses the points raised during the review process.

We look forward to receiving your revised manuscript.

Kind regards,

Andreas Buechner

Academic Editor

PLOS ONE

Journal Requirements:

Reviewers' comments:

Reviewer's Responses to Questions

**Comments to the Author**

1. Is the manuscript technically sound, and do the data support the conclusions?

Reviewer #1: Yes

Reviewer #2: Yes

2. Has the statistical analysis been performed appropriately and rigorously? 

Reviewer #1: Yes

Reviewer #2: N/A

3. Have the authors made all data underlying the findings in their manuscript fully available?

Reviewer #1: Yes

Reviewer #2: Yes

4. Is the manuscript presented in an intelligible fashion and written in standard English?

Reviewer #1: Yes

Reviewer #2: Yes

5. Review Comments to the Author

Reviewer #1: The article summarizes current evidence about facemasks’ impact on communication, examining effects on acoustics, intelligibility, and voice production. The study presents the results of original research, which to my knowledge have not been published elsewhere.

The problem is exhaustively illustrated in the introduction and a good overview of the type of existing masks is provided.

The methods are extensively described and appear rigorous. The methods follow high technical standards. Supplemental materials provide all data to better know the research and selection process.

The “result and discussion” section is subdivided into coherent subsections covering all the relevant aspects of the issue, providing a clear and complete state-of-the-art about the topic.

In the “perceptual outcomes of masks” section, I suggest focusing on the impact of facemasks on speech perception according to the hearing status of the listener, since this aspect has gained popularity in the scientific community and public opinion as well and deserves some adjunctive detail. On this latter point, I think that the work of Tofanelli et al. could add some valuable insights (Tofanelli M, Capriotti V, Gatto A, Boscolo-Rizzo P, Rizzo S, Tirelli G. COVID-19 and Deafness: Impact of Face Masks on Speech Perception. J Am Acad Audiol. 2022 Feb;33(2):98-104. doi: 10.1055/s-0041-1736577.)

The limitations of the study are clearly stated.

The conclusions are coherent with the shown data.

The figures are clear and easily understandable.

Minor points:

Line 440: What do you mean with Fivexx? I believe this is a typing mistake. If so, please correct it. Otherwise, the phrase needs clarification.

Lines 590-593: The repeated “xx” are probably typing mistakes, that need to be fixed.

In summary, given the large and heterogeneous field of investigation and the effective synthesis provided, I commend the authors for their great efforts in this study. The article is easily readable and written in comprehensible English. Once fixed the abovementioned small issues, I consider the work worthwhile to be published.

Reviewer #2: The authors present a literature review on the acoustic and perceptual effects of face masks on speech. It does a good job summarizing the key results for researchers who are new to the topic, though I would have liked to see more analysis of the conflicting results in the literature. For example, is there a pattern of methodological differences between studies that did or did not find significant impacts on speech intelligibility?

I would also appreciate more analysis of the studies' methodology overall (microphone placement, live talkers vs. mannequins, live listening vs. recordings, how to control for audio/visual variables, how to report attenuation data), since in this new research area there are not yet established best practices. Notably, many of the studies were conducted during lockdown conditions and had to conduct human-subjects experiments remotely.

Specific comments:

- The paper needs some copyediting. There are several "XX" placeholders and one "[CITE]" placeholder. The authors are inconsistent in their formatting of numbers in the text. The usual practice is to write out numbers less than ten, and to never start a sentence with a numeral.

- Line 126-134: This paragraph does not seem to fit into the surrounding paragraphs, which have already introduced respirators. The introduction is choppy overall, with some concepts introduced multiple times.

- Line 164: The purported benefits of transparent masks for listeners with hearing loss are controversial, as many of the reviewed articles point out. They can be helpful in some cases but not others, depending on the relative impact on auditory and visual cues for a particular mask and their relative importance for a particular listener. I would word this sentence more carefully.

- Line 522: As written, it appears that the subject of the verb "dampened" is "attenuation", which is not what the authors intend to convey.

- In the audio-visual section, it is not clear exactly what is being compared in each study. There are many variables that might be manipulated in such a study: presenting audio-only recordings with/without masks, audio-visual recordings with/without masks, masked audio imposed on unmasked video, unmasked audio on masked video, unmasked recordings processed to simulate masks, etc. It is unclear whether all the studies in this section were measuring the same thing, so please provide more detail.

- In the section on vocal intensity, what exactly is being measured? If intensity is the overall acoustic energy in the signal, we would not expect most masks to have a measurable effect. Most energy in speech is concentrated at low frequencies, where masks are mostly transparent; even if a mask completely blocked all high frequencies, that would barely change the total energy. Human talkers might increase their effort while wearing a mask, which would affect the overall intensity, but it is not clear if that is what is being studied by all the papers in this section.

- Table 3 is difficult to read. It is too cluttered to effectively convey information. Perhaps it needs to be broken up, or at last reformatted.

6. PLOS authors have the option to publish the peer review history of their article (what does this mean?). If published, this will include your full peer review and any attached files.

Reviewer #1: No

Reviewer #2: No

---

## [Author Response · Author response to Decision Letter 0]

6 Feb 2023

See also the uploaded Response to Reviewers document.

----

PONE-D-22-09426

Acoustic and perceptual impact of face masks on speech: A scoping review

PLOS ONE

Reviewers' comments:

Reviewer's Responses to Questions

Comments to the Author

1. Is the manuscript technically sound, and do the data support the conclusions?

Reviewer #1: Yes

Reviewer #2: Yes

2. Has the statistical analysis been performed appropriately and rigorously? 

Reviewer #1: Yes

Reviewer #2: N/A

3. Have the authors made all data underlying the findings in their manuscript fully available?

Reviewer #1: Yes

Reviewer #2: Yes

4. Is the manuscript presented in an intelligible fashion and written in standard English?

Reviewer #1: Yes

Reviewer #2: Yes

5. Review Comments to the Author

Reviewer #1: The article summarizes current evidence about facemasks’ impact on communication, examining effects on acoustics, intelligibility, and voice production. The study presents the results of original research, which to my knowledge have not been published elsewhere.

The problem is exhaustively illustrated in the introduction and a good overview of the type of existing masks is provided.

The methods are extensively described and appear rigorous. The methods follow high technical standards. Supplemental materials provide all data to better know the research and selection process.

The “result and discussion” section is subdivided into coherent subsections covering all the relevant aspects of the issue, providing a clear and complete state-of-the-art about the topic.

In the “perceptual outcomes of masks” section, I suggest focusing on the impact of facemasks on speech perception according to the hearing status of the listener, since this aspect has gained popularity in the scientific community and public opinion as well and deserves some adjunctive detail. On this latter point, I think that the work of Tofanelli et al. could add some valuable insights (Tofanelli M, Capriotti V, Gatto A, Boscolo-Rizzo P, Rizzo S, Tirelli G. COVID-19 and Deafness: Impact of Face Masks on Speech Perception. J Am Acad Audiol. 2022 Feb;33(2):98-104. doi: 10.1055/s-0041-1736577.)

RESPONSE: Thank you for this important point. We have added an initial subsection in the speech perception results reporting on the two articles that included participants with hearing loss. Given that this scoping review search took place in May 2021, we did not include the article you mentioned, though have made a point of being transparent in the introduction, discussion, and limitations that this review captures the state of a dynamic literature at a particular point in time (in particular on pages 5 & 45).

The limitations of the study are clearly stated.

The conclusions are coherent with the shown data.

The figures are clear and easily understandable.

Minor points:

Line 440: What do you mean with Fivexx? I believe this is a typing mistake. If so, please correct it. Otherwise, the phrase needs clarification.

RESPONSE: This was an artefact of our editing process. We have since reviewed and removed these instances throughout.

Lines 590-593: The repeated “xx” are probably typing mistakes, that need to be fixed.

RESPONSE: These have been removed. Thank you for bringing them to our attention.

In summary, given the large and heterogeneous field of investigation and the effective synthesis provided, I commend the authors for their great efforts in this study. The article is easily readable and written in comprehensible English. Once fixed the abovementioned small issues, I consider the work worthwhile to be published.

RESPONSE: Thank you for this feedback!

 

Reviewer #2: The authors present a literature review on the acoustic and perceptual effects of face masks on speech. It does a good job summarizing the key results for researchers who are new to the topic, though I would have liked to see more analysis of the conflicting results in the literature. For example, is there a pattern of methodological differences between studies that did or did not find significant impacts on speech intelligibility?

RESPONSE: Given that this is a scoping, rather than systematic review, a detailed synthesis of this nature is beyond the purpose of this review. However, we agree with your point that this is important information to consider. We have added information about the presence of noise and some speculation in our final subsection of the perceptual results (page 33).

I would also appreciate more analysis of the studies' methodology overall (microphone placement, live talkers vs. mannequins, live listening vs. recordings, how to control for audio/visual variables, how to report attenuation data), since in this new research area there are not yet established best practices. Notably, many of the studies were conducted during lockdown conditions and had to conduct human-subjects experiments remotely.

RESPONSE: Again, given the nature of a scoping review, we believe this level of synthesis is not necessarily appropriate or warranted. For this reason, we include the methodological details in the supplemental materials. We now also include an explicit mention of this type of limitation on page 46. 

Specific comments:

- The paper needs some copyediting. There are several "XX" placeholders and one "[CITE]" placeholder. The authors are inconsistent in their formatting of numbers in the text. The usual practice is to write out numbers less than ten, and to never start a sentence with a numeral.

RESPONSE: These have all been addressed; thank you for bringing these to our attention.

- Line 126-134: This paragraph does not seem to fit into the surrounding paragraphs, which have already introduced respirators. The introduction is choppy overall, with some concepts introduced multiple times.

RESPONSE: We agree with this comment. We have collapsed the “Role of masks” within this section. We believe this section is more cohesive now.

- Line 164: The purported benefits of transparent masks for listeners with hearing loss are controversial, as many of the reviewed articles point out. They can be helpful in some cases but not others, depending on the relative impact on auditory and visual cues for a particular mask and their relative importance for a particular listener. I would word this sentence more carefully.

RESPONSE: This sentence has been revised and worded more cautiously and with caveats.

- Line 522: As written, it appears that the subject of the verb "dampened" is "attenuation", which is not what the authors intend to convey.

RESPONSE: This has been fixed.

- In the audio-visual section, it is not clear exactly what is being compared in each study. There are many variables that might be manipulated in such a study: presenting audio-only recordings with/without masks, audio-visual recordings with/without masks, masked audio imposed on unmasked video, unmasked audio on masked video, unmasked recordings processed to simulate masks, etc. It is unclear whether all the studies in this section were measuring the same thing, so please provide more detail.

RESPONSE: Thank you for this comment. We have added substantially more detail in this section to make these differences clearer.

- In the section on vocal intensity, what exactly is being measured? If intensity is the overall acoustic energy in the signal, we would not expect most masks to have a measurable effect. Most energy in speech is concentrated at low frequencies, where masks are mostly transparent; even if a mask completely blocked all high frequencies, that would barely change the total energy. Human talkers might increase their effort while wearing a mask, which would affect the overall intensity, but it is not clear if that is what is being studied by all the papers in this section.

RESPONSE: We completely agree with your points, and this is exactly what we sought to characterize and is why we separated this section from attenuation. We have clarified your points in the first and last paragraphs in this subsection and added more detail on our reporting of some of the studies.

- Table 3 is difficult to read. It is too cluttered to effectively convey information. Perhaps it needs to be broken up, or at last reformatted.

RESPONSE: We have modified this into two tables, one reporting on acoustic outcomes and the other on perceptual outcomes.

6. PLOS authors have the option to publish the peer review history of their article (what does this mean?). If published, this will include your full peer review and any attached files.

Do you want your identity to be public for this peer review? For information about this choice, including consent withdrawal, please see our Privacy Policy.

Reviewer #1: No

Reviewer #2: No

RESPONSE: Done.

---

## [Decision Letter · Decision Letter 1]

14 Apr 2023

Acoustic and perceptual impact of face masks on speech: A scoping review

PONE-D-22-09426R1

Dear Dr. Knowles,

We’re pleased to inform you that your manuscript has been judged scientifically suitable for publication and will be formally accepted for publication once it meets all outstanding technical requirements.

Kind regards,

Andreas Buechner

Academic Editor

PLOS ONE

Additional Editor Comments (optional):

Reviewers' comments:

Reviewer's Responses to Questions

**Comments to the Author**

1. If the authors have adequately addressed your comments raised in a previous round of review and you feel that this manuscript is now acceptable for publication, you may indicate that here to bypass the “Comments to the Author” section, enter your conflict of interest statement in the “Confidential to Editor” section, and submit your "Accept" recommendation.

Reviewer #1: All comments have been addressed

Reviewer #2: All comments have been addressed

2. Is the manuscript technically sound, and do the data support the conclusions?

Reviewer #1: Yes

Reviewer #2: Yes

3. Has the statistical analysis been performed appropriately and rigorously? 

Reviewer #1: Yes

Reviewer #2: Yes

4. Have the authors made all data underlying the findings in their manuscript fully available?

Reviewer #1: Yes

Reviewer #2: Yes

5. Is the manuscript presented in an intelligible fashion and written in standard English?

Reviewer #1: Yes

Reviewer #2: Yes

6. Review Comments to the Author

Reviewer #1: (No Response)

Reviewer #2: (No Response)

7. PLOS authors have the option to publish the peer review history of their article (what does this mean?). If published, this will include your full peer review and any attached files.

Reviewer #1: No

Reviewer #2: No

---

## [Editor Report · Acceptance letter]

12 May 2023

PONE-D-22-09426R1 

Acoustic and perceptual impact of face masks on speech: A scoping review 

Dear Dr. Knowles:

I'm pleased to inform you that your manuscript has been deemed suitable for publication in PLOS ONE. Congratulations! Your manuscript is now with our production department. 

Kind regards, 

on behalf of

Andreas Buechner 

Academic Editor

PLOS ONE